# Happy just because. A cross-cultural study on subjective wellbeing in three Indigenous societies

Victoria Reyes-García[1,2,3]*, Sandrine Gallois[4], Aili Pyhälä[5,6], Isabel Díaz-Reviriego[7], Álvaro Fernández-Llamazares[6,8], Eric Galbraith[1,2,9], Sara Miñarro[2], Lucentezza Napitupulu[10]

1 Institució Catalana de Recerca i Estudis Avançats (ICREA), Barcelona, Spain, 2 Institut de Ciència i Tecnologia Ambientals, Universitat Autònoma de Barcelona, Barcelona, Spain, 3 Departament d'Antropologia Social i Cultural, Universitat Autònoma de Barcelona, Barcelona, Spain, 4 Faculty of Archaeology, Leiden University, Leiden, the Netherlands, 5 Global Development Studies, University of Helsinki, Helsinki, Finland, 6 Helsinki Institute of Sustainability Science (HELSUS), University of Helsinki, Helsinki, Finland, 7 Faculty of Sustainability, Social-Ecological Systems Institute (SESI), Leuphana University of Lüneburg, Lüneburg, Germany, 8 Faculty of Biological and Environmental Sciences, Global Change and Conservation (GCC), Organismal and Evolutionary Biology Research Programme, University of Helsinki, Helsinki, Finland, 9 Department of Earth and Planetary Sciences, McGill University, Montreal, Canada, 10 Department of Economics, Universitas Indonesia, Depok, Indonesia

* victoria.reyes@uab.cat

**Data Availability Statement:** All relevant data are within the manuscript and its Supporting information files.

## Abstract

While cross-cultural research on subjective well-being and its multiple drivers is growing, the study of happiness among Indigenous peoples continues to be under-represented in the literature. In this work, we measure life satisfaction through open-ended questionnaires to explore levels and drivers of subjective well-being among 474 adults in three Indigenous societies across the tropics: the Tsimane' in Bolivian lowland Amazonia, the Baka in southeastern Cameroon, and the Punan in Indonesian Borneo. We found that life satisfaction levels in the three studied societies are slightly above neutral, suggesting that most people in the sample consider themselves as moderately happy. We also found that respondents provided explanations mostly when their satisfaction with life was negative, as if moderate happiness was the normal state and explanations were only needed when reporting a different life satisfaction level due to some exceptionally good or bad occurrence. Finally, we also found that issues related to health and–to a lesser extent–social life were the more prominent explanations for life satisfaction. Our research not only highlights the importance to understand, appreciate and respect Indigenous peoples' own perspectives and insights on subjective well-being, but also suggests that the greatest gains in subjective well-being might be achieved by alleviating the factors that tend to make people unhappy.

**Funding:** This work was funded by the European Research Council (Grant Number FP7-261971-LEK to VRG). This work contributes to the ICTA-UAB Maria de Maeztu Unit of Excellence (CEX2019-000940-M) of the Ministerio de Ciencia, Innovación y Universidades, Spain. The funders had no role in study design, data collection and analysis, decision to publish, or preparation of the manuscript.

**Competing interests:** The authors have declared that no competing interests exist.

## Introduction

Happiness or subjective well-being, understood as the assemblage of mental and emotional states that emerge from the appraisal of life satisfaction and the balance of positive over negative emotions [1], has been a long-standing focus of scientific inquiry [2]. Happiness research has been conducted in a wide variety of scientific disciplines, including social psychology, gerontology, clinical research, and economics. This research has largely been based on the assumptions that the pursuit of personal happiness is universal [3], one of the most important values guiding individuals' life [4], and achievable by proactively taking control of one's life (see [5] for a review). Based on these assumptions, research on happiness has been largely dominated by attempts to measure–and compare–people's subjective well-being and to find its multiple correlates and drivers. Results suggest that 1) most people report being happy [6,7]; 2) both personal characteristics (e.g., age, marital status, income) and external factors (e.g., work satisfaction, governance, religion) shape happiness [2,8]; and 3) happiness itself is positively associated with desired societal outcomes, including health, productivity, and social life [9–11].

Yet, a growing body of cross-cultural research on subjective well-being highlights that both the concept of happiness and its drivers largely differ across cultures [12–14]. A critique to standard happiness research is that it has largely relied on a concept arising from modern individualist cultures that are neither historically nor cross-culturally representative [15]. Thus, far from being universal, the idea that happiness is an important value guiding individuals' lives appears to be strongly emphasized only in individualistic cultures [16]. Differently, in places where collectivistic values predominate, life is more firmly guided by social relations or by the need to have a purpose in life [17]. Furthermore, aversion to certain expressions of happiness is common in some cultures, and -for some-overtly expressing happiness is considered offensive and to attract bad luck [5,18]. Moreover, cross-cultural studies on subjective well-being have also shown that the drivers of happiness not only vary among individuals, but also across social groups. For example, among the several drivers of subjective well-being, income and material wealth have received broad research interest, under the assumption that they have the potential to fulfil universal needs, thus leading to happiness [19–21]. However, research in societies with low levels of monetization suggests the people can display high levels of subjective well-being without income [22] and that income might even have negative effects in terms of subjective well-being [23], thus raising issues on the absolute effect of income in happiness. It seems, then, that happiness research could continue to benefit from insights provided by cross-cultural analysis.

Particularly, the study of happiness among Indigenous peoples, a population largely under-represented in this literature, might bring relevant insights for our understanding of happiness. The few prior studies exploring subjective well-being among Indigenous societies have already highlighted that the concept and drivers of subjective well-being vary from one Indigenous society to another [6]. For example, among the Matsigenka in the Peruvian Amazon, the concept of well-being is fundamentally linked to health and productivity [24]. In contrast, for contemporary Inuit of the Arctic, happiness is defined and driven by family relations and participation in the social world [25]. Relations with other people, with nature, and with spiritual beings also seem to be central for the subjective well-being of Native Americans [26,27], Aboriginal Australians [28], or Amerindians [29]. For instance, among the Yorta Yorta Nation, Boonwurrung and Bangerang tribes of Victoria (Australia), "caring for country" and management of traditional lands is considered a key determinant of community well-being [30]. Similarly, among numerous Amazonian Indigenous peoples, the concepts of "the good life" or "living well" emphasize the importance of maintaining harmonious interpersonal

relationships, some of which extend to nonhumans and spirits [31–33]. Furthermore, some evidence indicates that changes in social relations or in relations with their local environments might erode substantially the subjective well-being of Indigenous peoples [23,27].

Our work contributes to cross-cultural research on happiness by exploring drivers of subjective well-being among 474 adults in three Indigenous societies across the tropics: the Tsimane' in Bolivian lowland Amazonia, the Baka in southeaster Cameroon, and the Punan in Indonesian Borneo. These three groups live in tropical rainforest environments and largely practice subsistence-based livelihood activities, including hunting, fishing, gathering, and agriculture [34]. All three societies maintain a number of traditional customs, norms, and beliefs related to collective community living and reciprocity, with their social and economic organization still largely based on kinship [34]. However, they are also increasingly dependent on cash to obtain certain commercial goods (i.e., machetes, mobile phones), or services (i.e., school fees and health treatments), for which the sale of forest products and sporadic employment play an increasingly important role in their socioeconomic systems. For each of these societies, we examine responses to questions aiming at measuring subjective well-being levels and variation in responses. We also examine the explanations given by individuals for particular self-reported levels of subjective well-being thus bringing participants' own perspectives on the drivers of their subjective well-being.

Beyond its contributions to cross-cultural research on happiness, this work also contributes to research on Indigenous people's well-being and its drivers. Most previous research on Indigenous peoples' well-being has focused on the status of objective and externally defined metrics and indicators (e.g., health, social, or economic indicators) [35], emphasizing the gap between Indigenous and non-Indigenous populations in attaining these indicators [36–38]. While pointing at critical issues, most of these studies overlook Indigenous peoples' subjective valuations of life satisfaction, including their own definitions of these indicators and their own culturally-specific explanations for them, for which this body or research potentially provides an incomplete and unbalanced view of Indigenous People's subjective well-being and its drivers [39]. Researchers have argued that, for many Indigenous societies, cultural attachment (e.g., use of Indigenous languages, participation in cultural activities, or spiritual practices) can be an important driver of well-being [40–43]. For example, aboriginal Australians in remote areas appear to be happier than those in non-remote areas, arguably as those in remote areas are more likely to speak their language, participate in hunting or gathering, and less likely to be discriminated against [44]. Similarly, research among Indigenous populations in the Arctic Alaska shows that wage income and job opportunities are negatively associated with their life satisfaction. Non-wage income instead, which allows time to be used for subsistence activities, is positively associated with life satisfaction [42]. Along these lines, legacies of reduced cultural engagement among Indigenous Peoples, as a result of colonialism (e.g., boarding schools, assimilation policies, language erasure), often lead to declines in peoples' physical and mental well-being, including feelings of shame and insignificance [45]. Together, these findings indicate that the conception of well-being for Indigenous populations might largely vary from the concept in Western societies. By bringing Indigenous People's perspectives on the drivers of subjective well-being, we help to provide a more accurate, complete and balanced overview of Indigenous Peoples' subjective well-being and its drivers.

## Case studies

The Tsimane' are a small-scale society of foragers and farmers in the Bolivian Amazon. They number ~ 14,000 people living in ~100 villages of commonly ~20 households per village, concentrated along rivers and logging roads [46]. Up until the late 1930s, the Tsimane' lived much

as they did prior to first contact with other sectors of society, maintaining a semi-nomadic and self-sufficient lifestyle. Their interactions with the Bolivian society have steadily increased since the 1940s and the Tsimane' are now largely settled in permanent villages with school facilities [46]. To this day, the Tsimane' continue to be mostly self-sufficient, their economy relying on slash-and-burn farming supplemented by hunting, fishing, and gathering. For the Tsimane', the main sources of income are the sale of crops (i.e., rice, maize and plantain) and the barter of palm thatch [47], although during the study period, 34.2% of the Tsimane' in the sample had no income from wage, sales, or barter [34]. Some Tsimane' also obtain income from wage labor in logging camps, cattle ranches, and in the homesteads of farmers.

The Baka are an Ubangian speaking group living in the rainforests of the Congo Basin. They are present in four different countries (Gabon, Congo, Central African Republic, and Cameroon), the majority being in Cameroon, with about 40,000 individuals [48]. Traditionally nomadic and living in forest camps, the Baka used to rely on hunting, fishing, gathering and bartering forest products for agricultural products with their Bantu-speaking neighbors [48]. Baka livelihoods, however, have largely changed over the last 50 years and at least the Baka in Cameroon are now mostly settled in villages along logging roads, have adopted agriculture, and have gained a higher access to formal school and health services [48]. Nowadays, Baka livelihoods mix foraging and farming, both in their own plot and in the plots of their neighbors. Sedentarization has also resulted in the introduction of the cash economy: merchants, logging and mining companies, and bush meat and non-timber forest products traders, are now present in the area inhabited by the Baka, increasing the availability of market products and Baka integration into the cash economy [49]. The Baka, nonetheless, continue to remain largely self-sufficient [34].

Finally, the Punan are a society of ~10,000 people, living in the mountainous interior of East Kalimantan, Indonesian Borneo [50]. Traditional Punan livelihoods were largely based on hunting, gathering of wild and semi domesticated plants, and bartering with the locally settled farmers [50]. Similar to the Tsimane' and the Baka, the Punan started to shift to a more sedentary lifestyle during the mid-1950s, a shift supported by government programs [50]. Although the Punan are no longer nomadic, they continue to engage in long travels and seasonal stays in the forest for hunting wild boars and gathering wild edibles and other forest products, a practice known as *mufut* [50,51]. The Punan mostly collect forest products for household consumption, but they also collect some products for sale, such as eaglewood or gaharu (*Aquilaria spp.*), head of hornbill, or bezoar stones. Nevertheless, nowadays, wage labor -including wage from work in government projects—provides significant and regular income for many Punan, even in remote villages [52].

## Methods

Data were collected in the framework of a larger cross-cultural project [34] and in coordination with national research institutions. We worked in two villages in each study site. A team composed of a researcher, a research assistant and a translator collected data in each village over 18 months of fieldwork (2012–2013). The first six months were devoted to learning the local language, building up trust with local people, training research assistants, collecting background information, pilot-testing methods, and developing protocols for systematic data collection. During the following 12 months, we collected data on individuals' subjective well-being and explanations of subjective well-being levels by means of quarterly interviews.

Before starting research, we obtained the required national research permits, and the agreement from the relevant political organizations. To work in the Tsimane' territory, we obtained written permission of the Great Tsimane' Council. To work among the Punan, we obtained

permission from RISTEK (Ministry of State for Research and Technology, Indonesia, SIP NO: 038/SIP/FRP/SM/II/2012). No specific permissions were required to work in the area where the Baka live. Before collecting data, we requested the oral Free Prior and Informed Consent of each person approached, noting that information would be anonymized, participation was strictly voluntary, and people could choose not to participate or withdraw from the study at any point, without any repercussions. According to the national data protection laws of Bolivia, Cameroon, and Indonesia, there are no cross-border transfer restrictions for non-personal data to the EU. We returned the results of our research to the communities in several ways, according to the participant communities' and local regional organizations' needs and demands (e.g., local workshops, radio programs, publications in local languages. See http://icta.uab.cat/etnoecologia/lek/). The research protocol received the approval of the ethics committee of the Autonomous University of Barcelona (CEEAH-04102010).

## Sample

We approached all the households in the studied villages and invited all adults (i.e., people ≥18 years) to participate in our study.

Overall, we collected information from 474 adults: 137 Tsimane' (70 men, 67 women); 225 Baka (97 men, 128 women); and112 Punan (59 men, 53 women). To assess variation in responses, we repeated the interviews four times, for which we have 1180 data points (406 from Tsimane', 463 from Baka, and 311 from Punan respondents).

## Data collection

We proxied individual subjective well-being through a single question on overall life satisfaction, defined as "people's explicit and conscious evaluation of their lives, often based on factors that the individual deems relevant" [53]. A self-report on life satisfaction is considered a valid indicator of quality of life and one of the most commonly used methods to assess subjective well-being [2,54,55]. Specifically, we asked: "Taking everything into consideration, would you say your life is: very bad (= 0), not good (= 1), fair (= 2), good (= 3), or very good (= 4)". We used a five-point scale with specific terms that generally capture life satisfaction levels because, after testing, this framing worked better than the most commonly used 5-item scale [53,56] and than the 10-point scale used in the World Value Survey (www.worldvaouesurvey.org) and Gallup World Poll [57]. To get insights into the processes underlying well-being judgements, immediately after this question, we asked respondents to explain to us the reasons for their response and, when an explanation was provided, we wrote down textual answers.

To assess within individual variation in responses, we repeated interviews every three months for a maximum of four times per individual. Since we could not interview all participants every quarter, the sample per individual varies. We collected only one response for 95 respondents, two for 110 respondents, three for 211 respondents, and four for the remaining 58 respondents.

## Data analysis

For most of the analysis, we grouped life satisfaction responses into three categories: *negative* (grouping 'very bad' and 'not good'), *neutral* (responses of 'fair'), and *positive* (grouping 'good' and 'very good'). We also grouped each individual data point in one of two general categories: *explained*, or data points for which people provided a textual explanation of their life satisfaction level, and *unexplained*, or data points for which people did not provide an explanation. To code the category *explained*, we used an inductive approach grouping similar textual explanations under the same 'explanation' category. Categories were further grouped into six general

overarching 'reasons', which include a) general responses, b) health, c) social life, d) self-sufficiency, e) income and material possessions, and e) other reasons (Table 1). The grouping was informed by the in-depth ethnographic knowledge of the researchers who had spent 18 months living with these societies. Some responses referred to more than one of our coded explanations. For example, the response "*I am sick and do not have enough food*" refers to "own physical health" and "basic needs" and was thus coded in both the category "health" and "self-sufficiency". Because some textual answers corresponded to two explanations, the total number of explanations (n = 813) is higher than the number of data points for which informants provided an explanation (n = 692).

We start the analysis by describing the average levels of life satisfaction across the three societies. Since we have more than one data point per informant, we calculated 1) the pooled and 2) informants' averages. We tested if differences across societies were statistically significant using a Kruskal-Wallis equality-of-populations rank test. We also explored variation across subjective well-being responses considering both intra-individual and temporal variation. To analyze intra-individual variation in levels of life satisfaction and their explanations, we used the subsample of respondents for which we have more than one data point. We calculated the percentage of informants who had reported the same life satisfaction level across different measurements versus the percentage of informants who had reported variations in life satisfaction levels. Temporal variation was analyzed by calculating averages per society for the different quarters of data collection. We then compared the distribution of *explained* and *unexplained* responses across our three life satisfaction categories (i.e., negative, neutral, and

**Table 1. Coding of textual explanations of individual life satisfaction levels.**

| Reason | Explanation | Examples of textual explanations |
|---|---|---|
| General | No specific reason given | No explanation given; "I'm just happy"; "I feel good"; "My heart tells me so" |
| | There is problem/no problem | "Because I am living well"; "No problems" |
| Health | General | "All the family is in good health"; "No one is sick" |
| | Own physical health | "Because I am healthy"; "Because I feel healthy" |
| | Others' physical health (includes death) | "Because my family is healthy"; "Because my baby is healthy"; "I feels sick" |
| | Own emotional health | "I worry about . . ."; "I am afraid to die"; "I think too much" |
| Social | General | "I have no problem with anyone"; "Problem with authorities" |
| | Household | "Fights with my wife because she is jealous"; "My family is happy"; "I got a new baby" |
| | Community | "People talk about me"; "There are gossips in the community"; "No one is angry" |
| | Socializing | "I miss my husband, who is away"; "I want to go visit my father"; "I am with my family"; "My cousin is visiting" |
| Self-sufficiency | Basic needs (food) | "My family has food"; "There is no meat to eat"; "We have eaten pork" |
| | Subsistence activities | "I have hunted a lot"; "I have plenty of fruits"; "The fish are about to arrive" |
| Income & material items | Money/material items | "I do not have a machete"; "I bought a motor"; "I have lost my canoe" |
| | Market-related activities | "I got a lot of money selling palm thatch"; "I am about to sell wood"; "Too much work"; "I can't work" |
| Other | Miscellaneous | "I drink alcohol and I am happy"; "I do not know how to read, and would like to go back to school"; "I feel blessed by God"; "I'm afraid of spirits"; "It is a nice day"; "My house was destroyed by the wind"; "The cold season is over" |

positive). In our final analysis, we only use the *explained* data points to explore the distribution of reasons provided by informants across life satisfaction levels. In all analyses, we differentiated among responses from the three studied societies. We used Stata 13 for all calculations.

## Results

### Life satisfaction levels and their variation

Across the three societies, the most common response to our question on life satisfaction level was "good", which represented 51.2% of Tsimane', 59.0% of Baka, and 46.6% of Punan responses (Fig 1). Moreover, in the three societies, there were more reports in the extreme positive (i.e., "very good") than in the extreme negative (i.e., "very bad") life satisfaction level (9.6% vs 1.2% among the Tsimane'; 15.6% vs. 0.7% among the Baka; and 7.4% vs. 4.2% among the Punan).

The average life satisfaction score across reports was 2.54 (SD = 0.97, n = 1180), falling somewhere between "fair" and "good". The average life satisfaction across informants was very similar, although variation was slightly lower (avg = 2.57, SD = 0.74, n = 474). Individual average life satisfaction scores were highest for the Baka (avg = 2.73, SD = 0.75, n = 225), followed by the Tsimane' (avg = 2.51, SD = 0.60; n = 137), and lowest for the Punan (avg = 2.31, SD = 0.80, n = 112). Differences in life satisfaction scores across societies were statistically significant using a Kruskal-Wallis equality-of-populations rank test (Chi square = 27.8, p = .0001, df = 2).

To explore variation in life satisfaction levels within individuals, we used the subsample of people who answered our survey more than once (n = 331). We found that life satisfaction scores did not vary much within the responses of an individual (Fig 2). For 49.5% of the subsample the score of their life satisfaction did not vary at all across quarters, and for an additional 25.1% life satisfaction only changed by one level. In other words, for 75% of the subsample, life satisfaction responses barely changed, if at all, during the study period. Fig 1 shows that, in each society, only about 20% of the responses fall into the category "Fair", which will not allow for significant changes in happiness levels. Nevertheless, less than 7% of the people in the subsample of people who answered the survey more than once reported changes in life satisfaction levels that varied by three or four levels (i.e., from"very bad" to"good"; or from"-bad" to"very good"; or vice versa), with a slightly higher number of cases reporting an increase rather than a decrease in life satisfaction. The qualitative analysis of responses to high variations shows that these were largely related to specific life events (e.g., widowhood, birth). Overall, Baka respondents experienced less changes in life satisfaction responses than Tsimane' and Punan respondents. More than 75% of Baka respondents reported no change or change by only one level in their life satisfaction, and only 4.1% reported changes by three levels.

The comparison of the distribution of responses shows important temporal variation in average levels of life satisfaction only among the Baka, for whom the average life satisfaction score was highest in the third quarter of data collection (avg = 2.9), coinciding with the time when Baka go to their forest camps for hunting and collecting honey and other non-timber forest products. In contrast, the average Baka life satisfaction score was almost 1 point lower (average = 2.06) during the first quarter of data collection, or the time for opening agricultural plots, when workload is at its peak and food is scarce. For the Punan and particularly for the Tsimane', temporal variation was less marked (Fig 3).

### Unexplained life satisfaction responses

Among the reports for which informants provided a life satisfaction evaluation, 488 (41.4%) were left *unexplained*. In other words, for almost half of the responses, informants did not

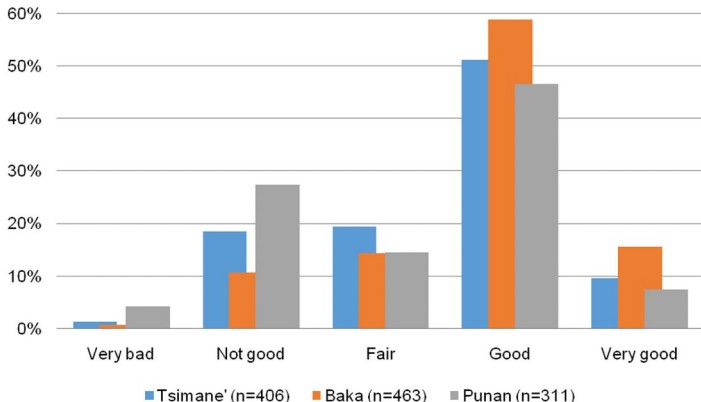

**Fig 1. Distribution of life satisfaction level responses, percentage by society.**

provide any reason for their answer. Reactions to that question varied from smiling or laughing at the enumerator to remaining silent, sometimes giving us an interrogative look, seemingly not really understanding the purpose of being asked the question. A handful of informants explicitly mentioned that they had never been asked nor thought about the issue before. *Unexplained* life satisfaction reports were common among the Baka, who left 240 (51.8%) of their life satisfaction reports unexplained, followed by the Punan, who did not explain 128 (41.2%) of the life satisfaction responses. *Unexplained* reports were lowest among the Tsimane', who left 120 (29.6%) of the life satisfaction responses unexplained.

Unexplained reports were not equally distributed across life satisfaction categories. In the three societies, the share of *unexplained* reports was larger for *positive* and *neutral* than for *negative* life satisfaction scores (Fig 4). For example, the Tsimane' left *unexplained* 37.3% of the 247 *positive* and 30.4% of the 79 *neutral* life satisfaction responses, but only 5% of the 80 *negative* life satisfaction responses. Similarly, the Baka left *unexplained* 56.2% of the 345 *positive* and 48.5% of the 66 *neutral*, but only 26.9% of the 52 *negative* life satisfaction responses. Following the same pattern, the Punan left *unexplained* 63.1% of the 168 *positive* and 33.3% of the 45 *neutral*, but only 7.4% of the 98 *negative* life satisfaction responses.

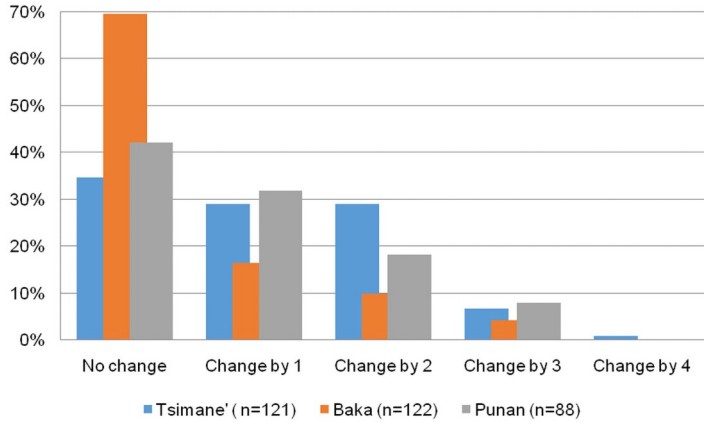

**Fig 2. Individual variation in life satisfaction score, percentage by society.**

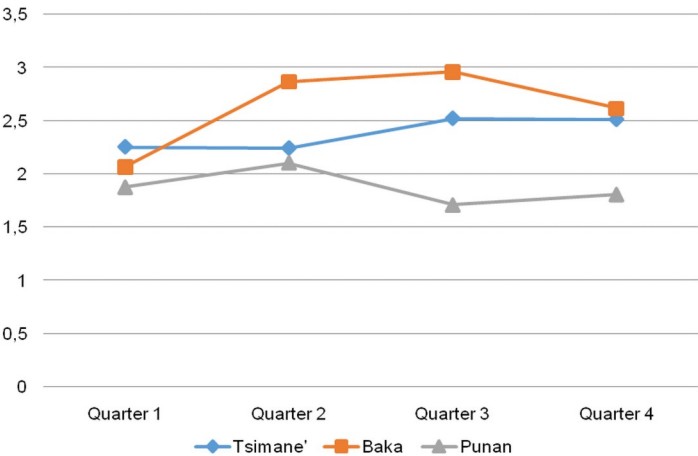

**Fig 3. Average life satisfaction score, by quarter of data collection and society.** Tsimane': Quarter 1, n = 90; Q2, n = 86; Q3, n = 69; Q4, n = 41; Baka: Quarter 1, n = 46; Q2, n = 15, Q3, n = 123, Q4, n = 39; Punan: Quarter 1, n = 48; Q2, n = 49; Q3, n = 55; Q4, n = 31.

### Reasons explaining life satisfaction responses

Overall, respondents provided explanations for 692 (58.6%) of their life satisfaction responses. The distribution of explanations across categories varied from one society to another. The Tsimane' provided 338 explanations for a total of 286 life satisfaction responses. Most of the explanations provided by the Tsimane' belong to three of the categories used: health (137 explanations, 40.5%), social life (77 explanations, 22.8%), and self-sufficiency (65 explanations, 19.2%). Explanations in the health category were evenly distributed among *positive*, *neutral*, and *negative* life satisfaction levels, whereas explanations in the social life and self-sufficiency categories were most often associated with *positive* than with *neutral* or *negative* life satisfaction levels (Fig 5).

Baka respondents provided 240 textual explanations for 223 life satisfaction reports. Most of the explanations provided by the Baka (143 explanations, 61%) fell into the category '*general*'. The second most common reason explaining Baka life satisfaction status was health (48 explanations, 20.0%). Explanations falling into the general category were mostly associated with *positive* life satisfaction levels, whereas explanations in the health category were evenly distributed among *positive*, *neutral*, and *negative* life satisfaction levels (Fig 5).

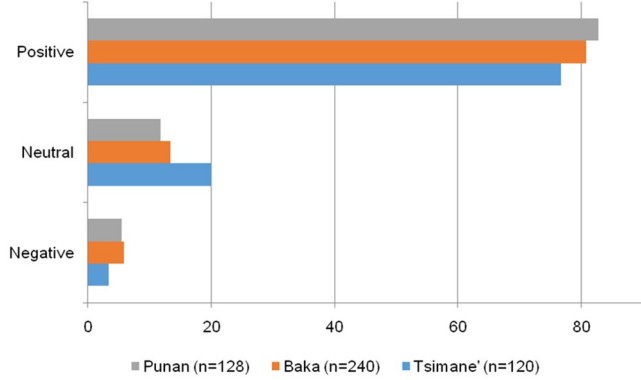

**Fig 4. Distribution of unexplained observations (n = 488), by life satisfaction categories and society.**

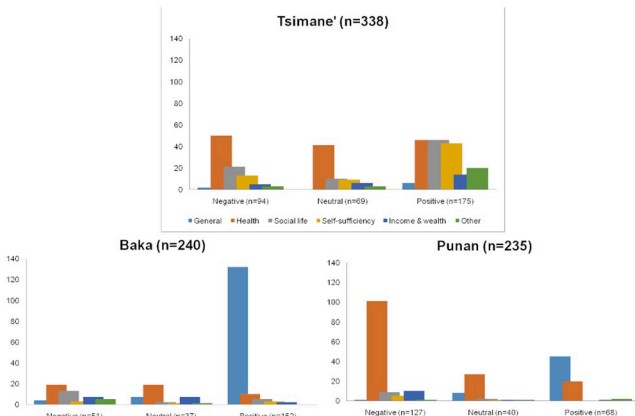

**Fig 5. Reasons explaining life satisfaction scores, by life satisfaction level and society.** Tsimane' n = 338; Baka n = 240; Punan n = 235.

The Punan provided 235 different explanations for 183 life satisfaction observations. Most explanations provided by the Punan relate to heath (n = 148, 63.0%), the second most commonly mentioned category being general (n = 54 explanations, 23.0%). For the Punan, health reasons are more frequently associated to *negative* life satisfaction levels (68.2% of responses in the health category), whereas explanations within the 'general' category largely correspond to *positive* life satisfaction levels (83.3% responses in the general category) (Fig 5).

## Discussion

Three main research findings derive from this work. First, life satisfaction levels in the three studied Indigenous societies are slightly above neutral, suggesting that most people in the sample identify themselves as moderately happy. Second, respondents often did not provide detailed explanations for their stated life satisfaction level, particularly when reporting positive life satisfaction. And finally, the explanations of life satisfaction varied across societies, although health and–to a lesser extent social life–were common among the three societies.

Before discussing the significance of these findings, we emphasize that they should be read with caution for two important reasons. First, our proxy for subjective well-being (i.e., life satisfaction) might suffer from measurement errors. Life satisfaction is increasingly being used in cross-cultural comparisons [56] because subjective evaluation allows for idiosyncratic reactions to different life circumstances [2]. However, as explained above, we used a variation of standard metrics of life satisfaction, which might potentially reduce the comparability of our results with research using other metrics. Second, our results should also be treated carefully because the expression of happiness itself is culturally patterned [58] and any cross-cultural research on subjective well-being faces the risk of exporting (or even imposing) Western concepts to non-Western cultures, potentially leading to invalid results [59]. We have deliberately tried to minimize this bias, by recording respondents' reactions and responses to our questions *verbatim*, but the bias might still be present. With these caveats in mind, we next discuss the main implications of our work.

First, in this work we found that, in the three studied Indigenous societies, life satisfaction levels are slightly above the average of the scale used, with few individuals reporting extreme (positive or negative) life satisfaction levels. The finding is consistent with previous research findings. For example, the average values found in our study are within a ± 1 point (in a 0–10

scale) of the average values found in the World Values Survey from the same countries in 2012. The Baka average value was higher than Cameroonian values (5.46 v. 4.24), whereas the Tsimane' and Punan average values were lower than Bolivian (5.02 vs. 6.02) and Indonesian (4.62 vs 5.37) average values, respectively. Life satisfaction levels reported here are also within the range of results from a major global analysis of industrial societies suggesting that most people are moderately happy [60] and within the range of results from an analysis done with a sample of people living in rural areas of developing countries [61]. Moreover, we found that life satisfaction levels were relatively stable throughout the year, large variations being explained by specific life events (e.g., death, birth). This pattern is also consistent with findings among other Indigenous societies [6] and with global personality research which argues that life satisfaction judgments are relatively stable over time because they are largely predicted by personality traits [62]. Overall, the alignment of our research results with previous research is noteworthy because it suggests that the measure used is relatively adequate to study subjective well-being among Indigenous societies. We note, however, that while our original question referred to overall subjective well-being, many of the explanations provided referred to recent events (e.g., "I got a new baby"). This raises the question on the importance of recent events shaping overall perceptions of well-being and calls for further research.

The second and probably most important finding of this work is that respondents often did not provide detailed explanations for their stated life satisfaction level. This was particularly the case when reporting positive life satisfaction level. In other words, participants found it easier to explain the reasons why their evaluations of life satisfaction were negative than to explain the reasons why their evaluations of life satisfaction were positive. We can think of several explanations to this finding. It is possible that, as it has been argued [63], humans react more strongly to negative than to positive stimuli, for which people might have a higher propensity to report negative emotions. It is also possible that the finding reflects a cultural bias in explicitly and actively expressing happiness, as it has been reported in certain cultures [5]. However, the finding could also be interpreted as suggesting that moderate happiness is, indeed, the normal human state, and most people would only have explanations for their life satisfaction level if they were above or below this state due to some exceptionally good or bad occurrence. Indeed, because moderate happiness facilitates positive social iterations without generating envy and jealousy, it might have been selected as an adaptive trait during our evolutionary history [60]. Research in the cultural evolution of happiness is needed to bring insights on the role of happiness in general, and moderate happiness in particular, in the evolution of our species.

The last finding of this work relates to variation in the reasons explaining life satisfaction from one society to another, and the particular role of health–and to a lesser extent social life– in the subjective well-being of the studied societies. Two important related considerations derive from this finding. On the one side, our findings add to a growing body of research suggesting that income and material wealth might not always be as important as other considerations, here health and social relations, in explaining subjective life satisfaction judgements [64,65]. It should be noted, however that this research has yet to address the degree to which this variation reflects specific situations, rather than cultural preferences. For example, while we found that a commonality in explaining life satisfaction among the three studied societies is the importance of health, we did not explore whether this was simply due to these societies having a low access to healthcare services, as is the case among many Indigenous societies [38], rather than a general cultural valuation of health. On the other side, this finding highlights Indigenous peoples' distinct perspectives on the drivers of subjective well-being. As mentioned, relatively little is known about Indigenous peoples own conceptualizations of happiness and the culturally-rooted determinants of their subjective well-being (see [42,44] for

some exceptions). In this study, we found that subjective well-being of the three Indigenous societies studies relates to health and social relations, not to income and material possessions, as seems to be the case in many Western societies [66]. Our findings, thus, dovetail with the emerging body of research that shows that Indigenous conceptualizations of well-being largely vary from the concept in industrial societies. The finding has important implications for policy-makers as it suggests that public policies for the well-being of Indigenous populations should address a variety of issues such as treatment of health problems or interactions with culture, communities, and nature. Most importantly, as the drivers of subjective well-being are culture-specific, Indigenous Peoples themselves should be at the center of policy planning in relation to their well-being, lifestyles, ways of making a living, and the future of their lands and waters. In many cases, this entails removing obstacles to their long-relationships with their lands and waters, and supporting Indigenous Peoples' efforts in sustaining their cultures, and their ways of knowing and being.

## Conclusion

As research on Indigenous conceptualizations of well-being accumulates from different communities around the world, we have an opportunity to carry out comparative analyses and identify commonalities on how well-being is understood and nurtured across Indigenous cultures. Our work highlights that, although culturally-grounded perspectives of happiness are as diverse as the Indigenous communities they emanate from, there are certain cross-cultural patterns worth investigating further.

Results on subjective well-being status and drivers in three Indigenous societies suggest that, in the absence of significant emotional stimuli (e.g., death, birth), most people tend to report being moderately happy. Respondents provided explanations mostly when their satisfaction with life was negative. The finding is important because it suggests that the greatest gains in subjective well-being might be achieved by alleviating the factors that make people unhappy, since they are clearly identifiable, and easily reported. Considering that non-Indigenous perspectives of well-being continue to guide the design and application of development policies in Indigenous lands [39], we call for stronger efforts to understand, appreciate and respect Indigenous determinants of subjective well-being, as perceived and reported by Indigenous peoples themselves.

## Supporting information

**S1 Protocol. Protocol for the questions on "wellbeing".**
(DOCX)

**S1 Data.**
(XLSX)

## Acknowledgments

We thank R. Duda and M. Guèze for collaboration in data collection. We thank our field teams for assistance during data collection: A. Ambassa, and E. Simpoh in Cameroon; V. Cuata, P. Pache, M. Pache, I. V. Sánchez, and S. Huditz in Bolivia; and S. Hadiwijaya and D. Suan in Indonesia, and X. Li and participants of the 1st LICCI Writing Workshop (FP7-771056-LICCI) for comments to a previous version of the article. We extend our deepest gratitude and respect to the Baka, Punan and Tsimane' societies, and to each of their individuals and villages, who so openly shared their friendship, happiness, and hospitality.

## Author Contributions

**Conceptualization:** Victoria Reyes-García, Aili Pyhälä.

**Formal analysis:** Victoria Reyes-García, Sandrine Gallois, Isabel Díaz-Reviriego, Álvaro Fernández-Llamazares, Eric Galbraith, Sara Miñarro.

**Funding acquisition:** Victoria Reyes-García.

**Investigation:** Victoria Reyes-García, Sandrine Gallois, Isabel Díaz-Reviriego, Álvaro Fernández-Llamazares, Lucentezza Napitupulu.

**Methodology:** Victoria Reyes-García, Aili Pyhälä.

**Project administration:** Victoria Reyes-García.

**Writing – original draft:** Victoria Reyes-García.

**Writing – review & editing:** Victoria Reyes-García, Sandrine Gallois, Aili Pyhälä, Isabel Díaz-Reviriego, Álvaro Fernández-Llamazares, Eric Galbraith, Sara Miñarro, Lucentezza Napitupulu.

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
