## [Decision Letter · Decision Letter 0]

9 Feb 2021

PONE-D-20-22427

Happy just because. A cross-cultural study on subjective wellbeing in three Indigenous societies

PLOS ONE

Dear Dr. Reyes-García,

Thank you for submitting your manuscript to PLOS ONE. After careful consideration, we feel that it has merit but does not fully meet PLOS ONE’s publication criteria as it currently stands. Therefore, we invite you to submit a revised version of the manuscript that addresses the points raised during the review process.

I would like to sincerely apologize for the long delay in reaching a decision on your submitted manuscript. There are multiple reasons for this, but I am happy to report that we have arrived at the end of this stage of the peer-review process. After receiving two reviews, we would like to recommend this piece for minor revision prior to final publication. 

Both peer-reviewers are very supportive of this manuscript, and offer some clear, specific guidance on revisions.  Reviewer 1's comments speak more to the framing of the piece and the literature review.  Reviewer 2's are tied closely to the results section. If you would please consider these suggested revisions and respond accordingly, we can commit to moving this process forward quickly.

Many thanks for your patience in this process, and we will look forward to seeing the revised manuscript.

We look forward to receiving your revised manuscript.

Kind regards,

Margaret Holland

Academic Editor

PLOS ONE

Journal Requirements:

2. Please include additional information regarding the survey or questionnaire used in the study and ensure that you have provided sufficient details that others could replicate the analyses. Specifically, please include a copy, in both the original languages used for each group and in English, as Supporting Information.

3. During our internal checks, the in-house editorial staff noted that you conducted research or obtained samples in another country. Please check the relevant national regulations and laws applying to foreign researchers and state whether you obtained the required permits and approvals. Please address this in your ethics statement in both the manuscript and submission information.

Reviewers' comments:

Reviewer's Responses to Questions

**Comments to the Author**

1. Is the manuscript technically sound, and do the data support the conclusions?

Reviewer #1: Yes

Reviewer #2: Yes

2. Has the statistical analysis been performed appropriately and rigorously? 

Reviewer #1: Yes

Reviewer #2: Yes

3. Have the authors made all data underlying the findings in their manuscript fully available?

Reviewer #1: Yes

Reviewer #2: No

4. Is the manuscript presented in an intelligible fashion and written in standard English?

Reviewer #1: Yes

Reviewer #2: Yes

5. Review Comments to the Author

Reviewer #1: This is a well written and very interesting piece. As someone with an intellectual background in Indigenous geographies, it is quite fascinating and dare I say, happiness-inducing to know that this type of work is being done with Indigenous communities. Your methodology and data collection are sound, and it was much appreciated on my end that the authors took a global view of indigeneity with their selections of communities in the manuscript.

I do have a few suggestions for revisions that I think will help this manuscript in the publication process.

1. The authors note that there is a literature surrounding the connection between happiness and well-being in communities, additionally noting that Indigenous perspectives are curiously missing within this literature. I would like to see the authors take a little more time in the manuscript to engage with this literature. It is clear that there is a gap, but what is the nature of the gap beyond an absence of Indigenous perspectives? What has been the historical arc of this literature, given that the authors tie in governmental policies of well-being into their analysis? The fact that this manuscript is compact and crisp in its writing is a strength, but I think that adding a bit more background in the beginning will only help to deepen the reader's understanding of what the authors are seeking to demonstrate in their analysis.

2. In a similar vein, I'd like to see a little bit more in the methods section, related to community relationship building and the work being done with communities. In an era where ethical research with Indigenous communities are becoming more and more of a requirement in the field, I think that some attention and care should be taken to further flesh out what this looks like in the authors' work.

3. Returning back to item #1, I'd also like to see the authors again address exactly how their work is filling existing gaps in the literature they cite. There is a mention of a gap, but I felt that I wanted to know more about the long term applicability of the work being done. If it is something that can be useful for governmental policies, what might that look like? Are there unanswered questions?

4. I think that there tends to be a move toward an idea of pan-Indigeneity towards the end of the paper, whereas the authors take care to note the unique local conditions of each community in the earlier part of the paper. I'd like to see this be more consistent throughout, all the way to the conclusion.

Reviewer #2: The authors present findings on the average happiness levels and what it means to be happy with data from 3 indigenous societies. They found most people were moderately happy and that explanations were more common for unhappiness rather than happiness. I found the data to be inherently interesting and applaud the authors for collecting both quantitative and qualitative data from diverse and often overlooked populations. The description of the study was nicely detailed and I found the conclusions discussed to be well-founded. However, I feel that the manuscript could be improved upon in the results section to more adequately represent and describe the results of the study. In particular, many of the analyses described are vague or do not include enough detail to follow what the authors actually did. While I believe that most of the important and interesting findings of the study can be adequately presented in descriptive statistics due to the qualitative nature of the data, for the few inferential statistics presented the authors need more detailed reporting.

The following are my main suggestions/comments:

When reporting the Kruskal-Wallis test of equality-of-populations the authors should include the corresponding chi-squared and df in addition to the p-value.

I am confused how the authors found “significant temporal variation in average levels of life satisfaction only among the Baka..” The authors report “temporal variation was analysed by calculating averages per society for the different quarters of data collection. We then analysed the distribution of explained and unexplained responses across our three life satisfaction categories” but they do not mention exactly how they calculated statistical significance. The authors should clearly report which tests were used and the corresponding results in greater detail.

The authors state that “less than 7% of the people in the sample reported changes in life satisfaction levels that varied by three or four levels” but what is the percent of people who initially reported an extreme response and then later changed by 3 or 4 levels? If over half of the respondents initially report their happiness towards the middle of the scale then there is a limit to how much they can deviate from their initial response. I’d rather see a percentage of the people who could actually change 3 or 4 levels and did so.

Along the lines of change in happiness, the authors should discuss how regression to the mean may play a role in the results they found.

Figure 5 is a bit hard to read given the limited responses for some of the categories. It may also be a bit misleading, as there are more explanations for positive responses just because more people were likely to respond positively to the happiness question. It might just be too difficult to include all of the information together in one chart. Perhaps breaking it apart into a pie chart displaying the breakdown of each categorical response for both positive and negative levels of happiness would be easier to interpret.

I thought the coding scheme did a suitable job of classifying the various responses given by the participants. I did notice though that many of the responses referred to a recent event as the explanation (e.g., “We have eaten pork,” “I got a new baby”) , while others were more general or over a longer time frame (e.g., “My family is healthy,” “People talk about me”). I wonder if the authors thought about coding the responses along some sort of timeframe (perhaps even just a dichotomous code of a recent event or not) and if this relates in any way to the level of happiness. For example, are recent events more likely to be described for less happy responses? If so, this would further strengthen the authors’ conclusion that happiness is considered the default state.

Minor edits/grammatical fixes:

Line 57: “highlights”

Line 133: -> ‘to this day’, (“until” implies this is no longer the case)

Line 158: As the Tsimane’ and the Baka -> Similar to (or like) the Tsimane’ and the Baka

Figure 3: Including sd or standard error bars on for the points would be helpful in understanding the variation in responses over time

6. PLOS authors have the option to publish the peer review history of their article (what does this mean?). If published, this will include your full peer review and any attached files.

Reviewer #1: **Yes: **Deondre A. Smiles, Ph.D.

Reviewer #2: No

---

## [Author Response · Author response to Decision Letter 0]

12 Apr 2021

Response to reviewers

Journal Requirements

 How addressed: We have formatted the manuscript following the requested instructions.

2. Please include additional information regarding the survey or questionnaire used in the study and ensure that you have provided sufficient details that others could replicate the analyses. Specifically, please include a copy, in both the original languages used for each group and in English, as Supporting Information.

 How addressed: The Supporting Information includes now a copy of the exact question used in this study in English, and a translation of the question to the languages used (i.e., Tsimane’, Baka and Punan).

3. During our internal checks, the in-house editorial staff noted that you conducted research or obtained samples in another country. Please check the relevant national regulations and laws applying to foreign researchers and state whether you obtained the required permits and approvals. Please address this in your ethics statement in both the manuscript and submission information.

 How addressed: National laws and regulations on data protection and transfer for Bolivia, Cameroon, and Indonesia are as follows:

Bolivia, Plurinational State of

 No data blocked

Legislation: Bill of Personal Data Protection and Ley general de Telecomunicaciones, Tecnologías de Información y Comunicación – Ley 167 de 08 agosto de 2011 (in Spanish)

No cross-border transfer restrictions for non-personal data to the EU.

Nothing in the Bill of Personal Data Protection isestablished concerning transfer.

Cameroon No data blocked

Legislation: No data protection legislation

No cross-border transfer restrictions for non-personal data to the EU.

Indonesia Qualified restriction for person data (Indonesia PPDES

Regulation).

No cross-border transfer restrictions for non-personal data to the EU.

We have added the following information in the manuscript.

Before starting research, we obtained the required national research permits, and the agreement from the relevant political organizations. To work in the Tsimane’ territory, we obtained written permission of the Great Tsimane’ Council. To work among the Punan, we obtained permission from RISTEK (Ministry of State for Research and Technology, Indonesia, SIP NO: 038/SIP/FRP/SM/II/2012). No specific permissions were required to work in the area where the Baka live. Before collecting data, we requested the oral Free Prior and Informed Consent of each person approached, noting that information would be anonymized, participation was strictly voluntary, and people could choose not to participate or withdraw from the study at any point, without any repercussions. According to the national data protection laws of Bolivia, Cameroon, and Indonesia, there are no cross-border transfer restrictions for non-personal data to the EU. We returned the results of our research to the communities in several ways, according to the participant communities’ and local regional organizations’ needs and demands (e.g., local workshops, radio programs, publications in local languages. See http://icta.uab.cat/etnoecologia/lek/). The research protocol received the approval of the ethics committee of the Autonomous University of Barcelona (CEEAH-04102010).

 How addressed: We have now made available the minimal anonymized data set necessary to replicate your study findings as Supporting Information file. 

Reviewer #1: 

General comment: This is a well written and very interesting piece. As someone with an intellectual background in Indigenous geographies, it is quite fascinating and dare I say, happiness-inducing to know that this type of work is being done with Indigenous communities. Your methodology and data collection are sound, and it was much appreciated on my end that the authors took a global view of indigeneity with their selections of communities in the manuscript.

 How addressed: We thank the reviewer for the overall positive assessment of our work.

I do have a few suggestions for revisions that I think will help this manuscript in the publication process.

Comment #1.1. The authors note that there is a literature surrounding the connection between happiness and well-being in communities, additionally noting that Indigenous perspectives are curiously missing within this literature. I would like to see the authors take a little more time in the manuscript to engage with this literature. It is clear that there is a gap, but what is the nature of the gap beyond an absence of Indigenous perspectives? What has been the historical arc of this literature, given that the authors tie in governmental policies of well-being into their analysis? The fact that this manuscript is compact and crisp in its writing is a strength, but I think that adding a bit more background in the beginning will only help to deepen the reader's understanding of what the authors are seeking to demonstrate in their analysis.

 How addressed:To address this comment, we have added some additional information to the last paragraph of the introduction, that now reads as follows

Beyond its contributions to cross-cultural research on happiness, this work also contributes to research on Indigenous People’s wellbeing and its drivers. Most previous research on Indigenous Peoples’ wellbeing has focused on the status of objective and externally defined metrics and indicators (e.g., health, social, or economic indicators) (35), emphasizing the gap between Indigenous and non-Indigenous populations in attaining these indicators (36–38). While pointing at critical issues, most of these studies overlook Indigenous peoples’ subjective valuations of life satisfaction, including their own definitions of these indicators and their own culturally-specific explanations for them, for which this research body potentially provides an incomplete and skewed overview of Indigenous peoples’ subjective wellbeing and its drivers (39). Researchers have argued that, for many Indigenous societies, cultural attachment (e.g., use of Indigenous languages, participation in cultural activities, or spiritual practices) can be an important driver of wellbeing, (Durie, Milroy, and Hunter 2009; Hossain and Lamb 2020; Wu 2020; King, Smith, and Gracey 2009). For example, aboriginal Australians in remote areas appear to be happier than those in non-remote areas, arguably as those in remote areas are more likely to speak their language, participate in hunting or gathering, and less likely to be discriminated against (Biddle 2014). Similarly, research among Indigenous populations in the Arctic Alaska shows that wage income and job opportunities are negatively associated with their life satisfaction. Non-wage income instead, which allows time to be used for subsistence activities, is positively associated with their life satisfaction (Wu 2020). Along these lines, legacies of reduced cultural engagement among Indigenous Peoples, as a result of colonialism (e.g., boarding schools, assimilation policies, language erasure), often lead to declines in peoples’ physical and mental well-being, including feelings of shame and insignificance (45). Together, these findings indicate that the conception of wellbeing for Indigenous populations might largely vary from the concept in Western societies. By bringing Indigenous People’s perspectives on the drivers of subjective wellbeing, we help to provide a more accurate, complete and balance overview of Indigenous Peoples’ subjective wellbeing and its drivers.

Comment #1.2. In a similar vein, I'd like to see a little bit more in the methods section, related to community relationship building and the work being done with communities. In an era where ethical research with Indigenous communities are becoming more and more of a requirement in the field, I think that some attention and care should be taken to further flesh out what this looks like in the authors' work.

 How addressed: We thank the reviewer for this important comment

Before starting research, we obtained the required national research permits, and the agreement from the relevant political organizations. To work in the Tsimane’ territory, we obtained written permission of the Great Tsimane’ Council. To work among the Punan, we obtained permission from RISTEK (Ministry of State for Research and Technology, Indonesia, SIP NO: 038/SIP/FRP/SM/II/2012). No specific permissions were required to work in the area where the Baka live. Before collecting data, we requested the oral Free Prior and Informed Consent of each person approached, noting that information would be anonymized, participation was strictly voluntary, and people could choose not to participate or withdraw from the study at any point, without any repercussions. According to the national data protection laws of Bolivia, Cameroon, and Indonesia, there are no cross-border transfer restrictions for non-personal data to the EU. We returned the results of our research to the communities in several ways, according to the participant communities’ and local regional organizations’ needs and demands (e.g., local workshops, radio programs, publications in local languages. See http://icta.uab.cat/etnoecologia/lek/). The research protocol received the approval of the ethics committee of the Autonomous University of Barcelona (CEEAH-04102010).

Comment #1.3. Returning back to item #1, I'd also like to see the authors again address exactly how their work is filling existing gaps in the literature they cite. There is a mention of a gap, but I felt that I wanted to know more about the long term applicability of the work being done. If it is something that can be useful for governmental policies, what might that look like? Are there unanswered questions?

 How addressed:To address this comment, in the last paragraph of the discussion section we have added the following sentences: 

On the other side, this finding highlights Indigenous Peoples’ distinct perspectives on the drivers of subjective wellbeing. As mentioned, relatively little is known about Indigenous Peoples’ own conceptualizations of happiness and the culturally-grounded determinants of their subjective wellbeing (see (42,44) for some exceptions). In this study, we found that subjective wellbeing of the three Indigenous societies studies relates to health and social relations, not to income and material possessions, as seems to be the case in many Western societies (60). Our findings, thus, dovetail with the emerging body of research that shows that Indigenous conceptualizations of well-being largely vary from the concept in industrial societies. These findings have important implications for policy-makers as they suggest that public policies for the wellbeing of Indigenous populations should address a variety of issues such as treatment of health problems or interactions with culture, communities, and nature. Most importantly, as the drivers of subjective wellbeing are culture-specific, Indigenous Peoples themselves should be at the center of policy planning in relation to their well-being, lifestyles, ways of making a living, and the future of their territories. In many cases, this entails removing obstacles to their long-relationships with their lands and waters, and supporting Indigenous Peoples' efforts in sustaining their cultures, and their ways of knowing and being.

Comment #1.4. I think that there tends to be a move toward an idea of pan-Indigeneity towards the end of the paper, whereas the authors take care to note the unique local conditions of each community in the earlier part of the paper. I'd like to see this be more consistent throughout, all the way to the conclusion.

 How addressed: To clarify our view on the topic, we have added the following paragraph at the beginning of the conclusion.

As research on Indigenous conceptualizations of well-being accumulates from different communities around the world, we have an opportunity to carry out comparative analyses and identify commonalities on how well-being is understood and nurtured across Indigenous cultures. Our work highlights that, although culturally-grounded perspectives of happiness are as diverse as the Indigenous communities they emanate from, there are certain cross-cultural patterns worth investigating further. 

Reviewer #2: 

The authors present findings on the average happiness levels and what it means to be happy with data from 3 indigenous societies. They found most people were moderately happy and that explanations were more common for unhappiness rather than happiness. I found the data to be inherently interesting and applaud the authors for collecting both quantitative and qualitative data from diverse and often overlooked populations. The description of the study was nicely detailed and I found the conclusions discussed to be well-founded. However, I feel that the manuscript could be improved upon in the results section to more adequately represent and describe the results of the study. In particular, many of the analyses described are vague or do not include enough detail to follow what the authors actually did. While I believe that most of the important and interesting findings of the study can be adequately presented in descriptive statistics due to the qualitative nature of the data, for the few inferential statistics presented the authors need more detailed reporting.

 How addressed: We thank the reviewer for the overall positive assessment of our work. Below we provide responses to the specific suggestions/comments.

The following are my main suggestions/comments:

Comment #2.1.When reporting the Kruskal-Wallis test of equality-of-populations the authors should include the corresponding chi-squared and df in addition to the p-value.

 How addressed: We have added the following clarification:

Differences in life satisfaction scores across societies were statistically significant using a Kruskal-Wallis equality-of-populations rank test (Chi square = 27.8, p=.0001, df = 2).

Comment #2.2.I am confused how the authors found “significant temporal variation in average levels of life satisfaction only among the Baka.” The authors report “temporal variation was analysed by calculating averages per society for the different quarters of data collection. We then analysed the distribution of explained and unexplained responses across our three life satisfaction categories” but they do not mention exactly how they calculated statistical significance. The authors should clearly report which tests were used and the corresponding results in greater detail.

 How addressed: The author is right in that our use of language is not very accurate here, as we did not conduct any statistical test, and our finding refers only to description of the data. To clarify the issue, we have rewritten the sentences in the methods and results section that now read:

[Methods]: Temporal variation was analyzed by calculating averages per society for the different quarters of data collection. We then compared the distribution of explained and unexplained responses across our three life satisfaction categories (i.e., negative, neutral, and positive). 

[Results]: The comparison of the distribution of responses shows important temporal variation in average levels of life satisfaction only among the Baka, for whom the average life satisfaction score was highest in the third quarter of data collection (avg=2.9), coinciding with the time when Baka go to their forest camps for hunting and collecting honey and other non-timber forest products.

Comment #2.3.The authors state that “less than 7% of the people in the sample reported changes in life satisfaction levels that varied by three or four levels” but what is the percent of people who initially reported an extreme response and then later changed by 3 or 4 levels? If over half of the respondents initially report their happiness towards the middle of the scale then there is a limit to how much they can deviate from their initial response. I’d rather see a percentage of the people who could actually change 3 or 4 levels and did so.

 How addressed: This is a good point raised by the reviewer, but with a non-easy way to present results. The sample of people to explore variation in life satisfaction levels within individuals included 331 respondents who answered the survey more than once, but not all them provided us responses in the 1st quarter. On the first paragraph of the results section and in Figure 1, we provide a detailed description of the distribution of responses across categories. Except for the respondents in the “Fair” category, which do not reach 20% in any of the three societies, all the other respondents could have moved by 3-4 levels. We have provided this explanation in the text, when we said

To explore variation in life satisfaction levels within individuals, we used the subsample of people who answered our survey more than once (n=331). We found that life satisfaction scores did not vary much within the responses of an individual (Fig 2). For 49.5% of the subsample the score of their life satisfaction did not vary at all across quarters, and for an additional 25.1% life satisfaction only changed by one level. In other words, for 75% of the sub-sample, life satisfaction responses barely changed, if at all, during the study period. Figure 1 shows that, in each society, only about 20% of the responses fall into the category “Fair”, which will not allow for significant changes in happiness levels. Nevertheless, less than 7% of the people in the subsample of people who answered the survey more than once reported changes in life satisfaction levels that varied by three or four levels (i.e., from ”very bad” to ”good”; or from ”bad” to ”very good”; or vice versa), with a slightly higher number of cases reporting an increase rather than a decrease in life satisfaction.

Comment #2.4. Along the lines of change in happiness, the authors should discuss how regression to the mean may play a role in the results they found.

 How addressed: We are afraid that we do not understand this comment made by the reviewer. We will be happy to address it in further revisions, should the reviewer provide a more detailed explanation. 

Comment #2.5. Figure 5 is a bit hard to read given the limited responses for some of the categories. It may also be a bit misleading, as there are more explanations for positive responses just because more people were likely to respond positively to the happiness question. It might just be too difficult to include all of the information together in one chart. Perhaps breaking it apart into a pie chart displaying the breakdown of each categorical response for both positive and negative levels of happiness would be easier to interpret.

 How addressed: Following this suggestion, we have changed the figure. The new figures (histograms) offer a better representation of the answers given for each level of happiness, for each society. 

Comment #2.6.I thought the coding scheme did a suitable job of classifying the various responses given by the participants. I did notice though that many of the responses referred to a recent event as the explanation (e.g., “We have eaten pork,” “I got a new baby”), while others were more general or over a longer time frame (e.g., “My family is healthy,” “People talk about me”). I wonder if the authors thought about coding the responses along some sort of timeframe (perhaps even just a dichotomous code of a recent event or not) and if this relates in any way to the level of happiness. For example, are recent events more likely to be described for less happy responses? If so, this would further strengthen the authors’ conclusion that happiness is considered the default state.

 How addressed: This is a very important point raised by the reviewer. We have checked our data to see whether we could add the code proposed (i.e., a dichotomous variable indicating whether the event cited in the explanation was recent or not). Unfortunately, we found that many of the observations do not have the required level of detail to conduct the proposed analysis. For example, for the response “I miss my wife” we do not know if this is a recent event (due to a trip, for instance), or a more long term event (as for example due to the death of the person). To address this comment of the reviewer, we have opted to add the following sentence at the end of the discussion section.

It is interesting to note that, while our original question referred to overall subjective wellbeing, many of the explanations provided referred to recent events (e.g., “I got a new baby”). This raises the question on the importance of recent events shaping overall perceptions of wellbeing and calls for further research.

Comment #2.7.

Minor edits/grammatical fixes:

Line 57: “highlights”

Line 133: -> ‘to this day’, (“until” implies this is no longer the case)

Line 158: As the Tsimane’ and the Baka -> Similar to (or like) the Tsimane’ and the Baka

Figure 3: Including sd or standard error bars on for the points would be helpful in understanding the variation in responses over time

 How addressed: We have made these minor fixes requested by the reviewer, except for the last one. We tried to introduce the standard error, but we think the new figure is less clear than the previous one. We are including it here for consideration, but in the manuscript we uploaded the previous one.

---

## [Editor Report · Decision Letter 1]

29 Apr 2021

Happy just because. A cross-cultural study on subjective well-being in three Indigenous societies

PONE-D-20-22427R1

Dear Dr. Reyes-García,

We’re pleased to inform you that your manuscript has been judged scientifically suitable for publication and will be formally accepted for publication once it meets all outstanding technical requirements.

Kind regards,

Margaret Holland

Academic Editor

PLOS ONE

Additional Editor Comments (optional):

I would like to thank the author team for their careful attention to the reviewer comments, and their efforts to incorporate the suggested revisions into an updated manuscript. I have reviewed these changes carefully, and think that this manuscript is now ready for publication. I congratulate the author team on their research process, and look forward to seeing this manuscript shared as an important contribution to the literature. Thank you as well for your patience in this process.
---

## [Editor Report · Acceptance letter]

4 May 2021

PONE-D-20-22427R1 

Happy just because. A cross-cultural study on subjective wellbeing in three Indigenous societies 

Dear Dr. Reyes-García:

I'm pleased to inform you that your manuscript has been deemed suitable for publication in PLOS ONE. Congratulations! Your manuscript is now with our production department. 

Kind regards, 

on behalf of

Dr. Margaret Holland 

Academic Editor

PLOS ONE